# Recombinant Expression and Purification of the Cyanobacterial Chaperone HtpG from *Synechococcus elongatus* PCC 7942

**DOI:** 10.3390/mps8050103

**Published:** 2025-09-06

**Authors:** Liqun Jiang, Ibrahim D. Boyenle, Nicolas Delaeter, Yanxin Liu

**Affiliations:** 1Institute for Bioscience and Biotechnology Research, University of Maryland, Rockville, MD 20850, USA; ljiang18@umd.edu (L.J.); iboyenle@umd.edu (I.D.B.);; 2Department of Chemistry and Biochemistry, University of Maryland, College Park, MD 20740, USA

**Keywords:** cyanobacterium, HtpG, Hsp90, chaperone, ATPase, protein homeostasis, stress response

## Abstract

The 90 kDa Heat Shock Protein (Hsp90) is an essential and highly conserved molecular chaperone that supports the folding and maturation of a diverse array of client proteins across prokaryotic and eukaryotic organisms. In bacteria, HtpG, the Hsp90 homolog, plays a central role in stress response and protein homeostasis, particularly under high-temperature and other stress conditions. Despite extensive studies on HtpG from *E. coli*, the biochemical properties and functional roles of cyanobacterial HtpG remain poorly characterized. Here, we focus on HtpG from the cyanobacterium *Synechococcus elongatus* PCC 7942 (seHtpG), a model organism for photosynthesis and circadian rhythm research. We developed a method for the overexpression and purification of seHtpG in *E. coli*, achieving high purity and yield suitable for biochemical and structural studies. Biophysical and biochemical assays show that seHtpG forms dimers and hydrolyzes ATP at a rate of 1.9 ATP/min, 4-fold faster than that of *E. coli* HtpG. This work establishes seHtpG as a model for studying the roles of HtpG in cyanobacterial protein homeostasis, photosynthesis, and stress response, enabling further exploration of cyanobacterial Hsp90 in ecosystem dynamics and biotechnological applications.

## 1. Introduction

The 90 kDa Heat Shock Protein (Hsp90) is a highly conserved and ubiquitous molecular chaperone essential for the folding and maturation of a wide range of client proteins in both prokaryotic and eukaryotic cells [1,2]. In humans, Hsp90 stabilizes many oncogenic proteins—including kinases, hormone receptors, and transcription factors—that are crucial for signal transduction and cellular regulation [3]. In bacteria, HtpG (high-temperature protein G) serves as the Hsp90 homolog and is central to bacterial stress responses and protein homeostasis, especially under high-temperature or other stress conditions [4,5]. Both *E. coli* HtpG and yeast Hsp90s have been extensively studied as model systems for the Hsp90 family, while the therapeutic potential of human Hsp90s has driven significant research interest [1,6]. In addition to its presence in *E. coli*, the gene encoding HtpG has been identified in cyanobacteria [7,8]. However, the biochemical properties of cyanobacterial HtpG, and its roles in photosynthesis and stress response, remain poorly understood.

Cyanobacteria are prokaryotic, oxygenic phototrophs with a global distribution, inhabiting nearly every ecological niche on Earth, from tropical soils to the icy extremes of Antarctica, as well as extreme environments such as bare rocks and polluted wastewater [9]. Through oxygenic photosynthesis, they are central drivers of ecosystem nutrient cycling and play a fundamental role in global carbon and nitrogen fixation [10]. Beyond their ecological importance, cyanobacteria are prolific producers of bioactive compounds, supporting applications in bioremediation, biofertilizers, biofuels, food additives, and pigments [11,12].

Their remarkable ecological success is attributed in part to their ability to synthesize protective metabolites and stress-responsive proteins—including compatible solutes, antioxidants, and molecular chaperones—that buffer against fluctuations in temperature, salinity, and oxidative conditions. Among these, the molecular chaperone HtpG emerges as a particularly versatile factor, functioning as an adaptive trait that enhances resilience to diverse stresses [7,8,13,14]. By stabilizing proteins during cold shock, reinforcing defenses against oxidative stress, and maintaining proteostasis across shifting environments, HtpG exemplifies the molecular strategies that enable cyanobacteria to persist and thrive in ecologically and climatically diverse habitats [7].

In this study, we investigate the cyanobacterial HtpG from *Synechococcus elongatus* PCC 7942, which is a well-established model organism widely used for research on prokaryotic photosynthesis, circadian rhythms, and as a biotechnological platform [15,16]. To investigate the structure and function of seHtpG, one must recombinantly overexpress and purify seHtpG to homogeneity. The expression and purification of *E coli* HtpG and other Hsp90 homologs has been described previously, but only as brief methodological notes [17,18,19]. Although seHtpG has been expressed and purified previously, the detailed procedures were not reported, and contamination was observed in the final protein preparation [7,8]. Here, we report the successful production and purification of recombinant seHtpG in *E. coli* using fast protein liquid chromatography (FPLC) with a sequential workflow of nickel-affinity, anion-exchange, and size-exclusion chromatography. We further characterize its oligomerization state and ATPase activity, laying a foundation for future studies on the role of HtpG in protein homeostasis, photosynthesis, and stress response in cyanobacteria.

## 2. Materials and Methods

### 2.1. Sequence Alignments

The sequence alignments to show the identity and similarity between Hsp90 homologs were analyzed using the Clustal Omega program (http://www.ebi.ac.uk/Tools/msa/, accessed on 20 August 2024). The Hsp90 sequences were obtained from Uniprot database and include the following Hsp90s homologs with abbreviations and Uniprot IDs shown in the parentheses: HtpG from *Synechococcus elongatus* PCC 7942 (seHtpG, Q79N42), HtpG from *Synechococcus* sp. PCC 7002 (ssHtpG, B1XQJ4), HtpG from *E. coli* (ecHtpG, P0A6Z3), Hsc82 from *Saccharomyces cerevisiae* (scHsc82, P15108), α- and β-isoforms of cytosolic Hsp90 from human (hsHsp90α, P07900; hsHsp90β, P08238), mitochondrial Hsp90 from human (hsTrap1, Q12931), endoplasmic reticulum Hsp90 from human (hsGrp94, P14625), cytosolic Hsp90 from *Arabidopsis thaliana* (atHsp901, P27323), chloroplastic Hsp90 from *Arabidopsis thaliana* (atHsp905, Q9SIF2), and Hsp90 form *Chlamydomonas reinhardtii* (crHsp90, A8J1U1). The detailed species information is provided in Appendix A. The phylogenetic tree was constructed using the neighbor-joining method on Mega 11 [20] (https://www.megasoftware.net/, accessed on 20 August 2024).

### 2.2. Protein Expression

The DNA sequence of seHtpG was codon-optimized for expression in *E. coli* and cloned into a pET28a expression vector by Twist Bioscience, incorporating an N-terminal 6xHis tag followed by a TEV protease cleavage site. Protein expression was carried out in *E. coli* BL21 (DE3)-RIL and Rosetta (DE3) cells, which were grown at 37 °C with shaking at 150 rpm to an OD600 of 0.4–0.6. For induction, the temperature was maintained at 37 °C or reduced to 16 °C, followed by the addition of 0.5 mM IPTG. Note that IPTG was added for the low-temperature induction only after the culture had cooled down to 16 °C. Protein expression proceeded for 3 h at 37 °C or overnight at 16 °C with continuous shaking at 150 rpm.

### 2.3. Protein Purification

*E. coli* cells from 1 L cultures were harvested by centrifugation and resuspended in 30 mL buffer containing 40 mM Tris·HCl (pH 7.5), 480 mM KCl, 20 mM imidazole, and 1 mM β-mercaptoethanol. Lysis was performed by homogenization and sonication (Misonix Sonicator 3000, Qsonica, Newtown, CT, USA). For sonication, the resuspended cells were placed in an ice bath to prevent protein denaturation from heat). A power of 45 W is used with 2 s ON/2 s OFF pulse sequence. The resulting cell lysates were subjected to centrifugation (30,000× *g* for 30 min) and filtration through a 0.45 μm pore size membrane. Initial purification of seHtpG was conducted using Ni-NTA affinity chromatography with elution buffer composed of 40 mM Tris·HCl (pH 7.5), 150 mM KCl, 300 mM imidazole, and 1 mM β-mercaptoethanol. The elution from the Ni-NTA column was dialyzed overnight to remove imidazole, with TEV protease added to cleave the 6xHis tag. The buffer used for dialysis contains 40 mM Tris·HCl (pH 7.5), 150 mM KCl, and 1 mM Dithiothreitol. Further purification was achieved on AKTA Pure chromatography system from Cytiva using ion exchange chromatography (MonoQ 10/100 GL column, GE Life Sciences, Wilmington, DE, USA) and size exclusion chromatography (SEC) (Superdex 200 16/60 column, GE Life Sciences, Wilmington, DE, USA). The ion exchange chromatography was performed at a flow rate at 3 mL/min. The KCl salt concentration gradient from 25 mM to 45 mM was created by mixing buffer B (40 mM Tris·HCl pH 7.5, 1 M KCl, and 1 mM Dithiothreitol) with buffer A (40 mM Tris·HCl pH 7.5, and 1 mM Dithiothreitol. The buffer for SEC contains 40 mM HEPES-KOH (pH 7.5), 150 mM KCl, and 0.5 mM TCEP. Elution profiles were monitored as absorbance at 280 nm, and the efficiency of each purification step was assessed via 4–12% SDS-PAGE (Invitrogen, Thermo Fisher Scientific, Sunnyvale, CA, USA). Protein concentrations were determined spectroscopically using the molar extinction coefficient (ε), calculated based on the amino acid sequence. Proteins in SEC buffer were flash-frozen in liquid nitrogen for long-term storage.

### 2.4. Analytical Ultracentrifugation

Analytical ultracentrifugation (AUC) was conducted using a Beckman Optima XL-A analytical ultracentrifuge. Prior to AUC runs, seHtpG was dialyzed overnight at 4 °C in 40 mM HEPES–KOH (pH 7.5), 150 mM KCl, and 1 mM β-mercaptoethanol. AUC experiments were performed at 45,000 rpm (AN-60Ti rotor) for 8–10 h at 20 °C, with absorbance monitored at 280 nm. Scans were collected at approximately 60 s intervals with a radial step size of 0.001 cm. Sedimentation velocity analysis was carried out using SEDFIT/SEDPHAT (16-1c, NIH) software to determine the standard sedimentation coefficients (s20, w). Buffer viscosity (η = 1.0254 × 10^−2^ poise), buffer density (ρ = 1.00847 g/mL), and the partial-specific volume of seHtpG (0.7300 mL/g at 20 °C, as a dimer) were calculated using the Sednterp program (http://philo.rasmb.org/software/sednterp/, assessed on 25 May 2023).

### 2.5. Steady-State Enzymatic Coupled ATPase Assay

The steady-state ATP hydrolysis rate of seHtpG was measured using an enzymatic coupled ATPase assay [21]. The assay is based on a reaction in which the regeneration of hydrolyzed ATP is coupled to the oxidation of NADH. NADH exhibits an absorption peak at a wavelength of 340 nm. The final assay mixture contained 40 mM HEPES-KOH (pH 7.5), 150 mM KCl, 1 mM MgCl_2_, 1 mM ATP, 0.6 mM NADH, 1 mM phosphoenol pyruvate (PEP), 25 U/mL pyruvate kinase (PK) (Sigma), 25 U/mL lactate dehydrogenase (LDH) (Sigma), and 4–20 µM seHtpG. A seHtpG-free group was used as negative control. The reactions were initiated by the adding of ATP, and then NADH consumption was monitored by a decrease in absorbance at 340 nm on a Spectramax iD5 plate reader (Molecular Devices, LLC, San Jose, CA, USA), with ADP release rates calculated using a standard curve over 0–650 µM ADP.

## 3. Results and Discussion

### 3.1. Sequence Analysis

The genome of *S. elongatus* PCC 7942 contains a single gene encoding HtpG (Uniprot ID Q79N42; referred to hereafter as seHtpG). Sequence alignments between seHtpG and other Hsp90 family members reveal high sequence conservation (Figure 1). seHtpG shares higher than 50% of its identity with other cyanobacteria and around 30% of its identity with homologies from other organisms, as shown in Figure 1B, consist with early studies [7]. Structurally, seHtpG comprises 638 amino acids that can be divided into three domains, N-terminal domain (NTD, residue 1–214), middle domain (MD, residue 215–476) and C-terminal domain (CTD, residue 477–638), based on sequence alignments with *E.coli* HtpG [22]. There is a conserved ATP-binding pocket in its NTD, supporting its predicted ATPase activity. A conserved dimerization interface in CTD suggests that seHtpG forms a dimer, while the MD, connecting the NTD and CTD, is also conserved.

To explore the evolution of cyanobacterial HtpG, we analyzed the protein sequence of HtpG from *S. elongatus* PCC 7942 (seHtpG, UniProt ID: Q79N42) in comparison with orthologs from *Synechococcus* sp. PCC 7002, *Chlamydomonas reinhardtii* (microalgae), *Arabidopsis thaliana* (plant), *E. coli* (bacteria), *Saccharomyces cerevisiae* (yeast), and *Homo sapiens* (human). Eukaryotic organisms typically harbor multiple paralogs of Hsp90. For example, humans have four distinct Hsp90 genes: two cytosolic isoforms and two organelle-specific isoforms, one localized to the endoplasmic reticulum and the other to mitochondria. Note that the genome of *S. elongatus* PCC 7942 contains only a single copy of the *htpg* gene (NCBI: ABB57843.1).

Sequence alignment reveals a high degree of conservation, particularly within the cyanobacterial species. The pairwise sequence identities among the representative Hsp90 sequences are shown in Figure 1. Like other Hsp90s, seHtpG includes three conserved domains: NTD, MD and CTD. However, seHtpG lacks the charged linker, as well as the MEEVD motif at the C-terminus, both of which are characteristic of cytosolic Hsp90s in eukaryotes (the top and bottom fragments in Figure 1A). This absence of the charged linker and MEEVD motif is also observed in *E. coli* HtpG and the human mitochondrial Hsp90, Trap1. Notably, for all the cyanobacterial HtpG sequences we checked (*n* > 20), we identified a unique 10–30 residue insertion (E391 to T408 in seHtpG, the middle fragment in Figure 1A) between the large and small MD segments, a feature absents in non-cyanobacterial species. Although the sequence and length of this insertion vary, its presence is conserved across cyanobacteria. The structural and functional implications of this insertion are yet to be determined.

Photosynthetic cyanobacteria are widely regarded as the evolutionary origin of chloroplasts in eukaryotic plant cells [23]. Consequently, we anticipated that seHtpG would be phylogenetically close to Hsp90 homologs in algae or higher plants, particularly the chloroplastic Hsp90. However, seHtpG displays its closest phylogenetic relationship with the human mitochondrial Hsp90, Trap1 (Figure 1B). It is widely accepted that Mitochondria, like chloroplasts, have originated from an ancient endosymbiotic event [24,25]. These findings suggest a potential common ancestry between the cyanobacterium-like ancestor of chloroplasts and the bacterium-like ancestor of mitochondria, indicating possible shared evolutionary origins between these organelles.

### 3.2. Recombinant Overexpression of seHtpG in E. coli

To optimize overexpression of recombinant seHtpG in *E. coli*, we tested various *E. coli* strains as hosts, IPTG induction temperature, and induction duration. Results of protein expression were analyzed on cell lysates by SDS-PAGE with Coomassie staining (Figure 2). A prominent band between 65 and 80 kDa indicates successful overexpression of seHtpG, which has a theoretical molecular weight of 72.6 kDa for the wild-type protein, or 76 kDa for the fusion protein including the 6xHis tag, TEV cleavage site, and linkers. We evaluated two *E. coli* strains, Rosetta (DE3) and BL21 (DE3)-RIL, for seHtpG expression. Both strains supported robust overexpression of seHtpG without significant differences by visual evaluation, but Rosetta (DE3) showed slightly thicker target band (Figure 2). The overexpression of seHtpG has been observed at high temperature of 37 °C, as well as low temperature of 16 °C. High protein yield indicated as obvious thickest band at target molecular weight was achieved under both induction conditions: 3 h at 37 °C or overnight at 16 °C following IPTG addition. Our findings indicate that cyanobacterial HtpG, as exemplified by seHtpG, is well-suited for efficient recombinant overexpression in *E. coli*.

### 3.3. Purification of Recombinant seHtpG

The purification of recombinant seHtpG was conducted through a three-step process. First, the overexpressed protein was subjected to Ni-NTA affinity chromatography, followed by anion exchange chromatography and then size-exclusion chromatography. The N-terminal 6xHis tag was removed after affinity purification via cleavage with TEV protease. The elution profile from anion exchange chromatography is shown in Figure 3A. The protein eluted as a single peak at a conductivity of 31 mS/cm, which corresponds to an approximate salt concentration of 300 mM KCl. This finding indicates that the surface of seHtpG is negatively charged, consistent with its isoelectric point (pI ≈ 4.9). Furthermore, the protein’s propensity to bind to the anion exchange column is likely enhanced by its dimerization, as confirmed by subsequent size-exclusion chromatography and analytical ultracentrifugation (AUC).

To further improve the protein purity, the final step in the purification of seHtpG involved SEC, which also served the purpose of buffer exchange and putting the protein into the storage buffer (40 mM HEPES pH 7.5, 150 mM KCl, 0.5 mM TCEP). The size-exclusion chromatography elution profile displayed a single peak corresponding to seHtpG (Figure 3C). Consistently, SDS-PAGE analysis revealed a single band corresponding to seHtpG, with no detectable contaminating proteins (Figure 3D). The observed molecular weight on SDS-PAGE fell between 65 kDa and 80 kDa, consistent with the theoretical molecular weight of 72.6 kDa for monomeric seHtpG. The band position on gel is consistent with the observation of the band located slightly above 67 kDa by Sato et al. [8]. The size-exclusion chromatography peak eluted at 65 mL with a flow rate of 0.5 mL/min (Figure 3C), corresponding to a molecular weight close to that of the 158 kDa IgG standard provided by the column manufacturer (GE Life Sciences). This gel-filtration result indicates that recombinant seHtpG exists predominantly as a dimer, consistent with other Hsp90 homologs. From 1 L *E. coli* culture in Terrific Broth medium, around 20 mg seHtpG was obtained following the three-step purification process. This yield is much higher than the expression and purification Hsp90 homolog, like 1 mg human Hsp90β of from 100 mL culture [26], 0.5 mg human Hsp90α of 100 mL bacterial culture [27]. Our 3-step purification protocol also improved the seHtpG purity compared to previous attempts for purifying the same protein [8].

### 3.4. seHtpG Forms a Dimer

To assess the oligomeric state of purified seHtpG, we performed analytical ultracentrifugation (AUC) as described in the Methods section. The sedimentation velocity experiment confirmed that seHtpG is dimeric, with a sedimentation coefficient of sw = 5.458 S, sw(20, w) = 5.743 S, corresponding to an approximate molecular weight of 130 kDa (Figure 4). Other Hsp90 orthologs exhibit similar sedimentation coefficients, including 5.96 S for LbHsp90 [28], 6.10 S for porcine brain Hsp90 [29], 6.0 S for human mitochondrial Trap1 [19], and 5.6 S for *E. coli* HtpG [30]. The AUC results for seHtpG being a dimer are consistent with findings from size-exclusion chromatography.

### 3.5. ATPase Activity of Recombinant seHtpG

We confirmed that the purified recombinant seHtpG is active by measuring its ATPase activity. Members of the Hsp90 family are characterized by intrinsic ATPase activity [31]. To investigate the biochemical function of seHtpG, enzyme-coupled ATPase assays were conducted at various seHtpG concentrations in the presence of 1 mM ATP and 1 mM Mg^2+^. ATPase activity was assessed by monitoring the rate of NADH consumption, which is coupled to ADP release from ATP hydrolysis. In detail, the assay contains three sequential reactions: (1) ATP is hydrolyzed by seHtpG where ADP and inorganic phosphate are produced; (2) released ADP is regenerated to ATP in the presence of PK by which PEP is converted to pyruvate; (3) the pyruvate is reduced to lactate by LDH together with oxidation of NADH to NAD+. The decrease in NADH absorbance at 340 nm (Figure 5A) indicates that seHtpG is functional and actively hydrolyzes ATP. The hydrolysis rate was found to depend linearly on the concentration of seHtpG (Figure 5B). To determine the average ATP hydrolysis rate, the ADP release rate was normalized to the seHtpG monomer concentration, then averaged across three different concentrations. The final normalized ATP hydrolysis rate was 1.92 ± 0.26 ATP/min at 37 °C, which is higher than the reported ATPase activities of 0.4 ATP/min for Hsp82 and 0.48 ATP/min for *E. coli* HtpG [32], 0.8 ATP/min for *Mycobacterium tuberculosis* HtpG [33] at the same temperature.

## 4. Discussion

In this study, we established a robust workflow for the recombinant expression and purification of the cyanobacterial chaperone HtpG from *Synechococcus elongatus* PCC 7942 (seHtpG). Our results demonstrate that seHtpG can be overexpressed in *E. coli* and purified to high homogeneity using a three-step chromatographic procedure. Compared with previous reports [7,8], this protocol significantly improves both yield and purity, enabling downstream biochemical and structural analyses. Such methodological advances are essential for the systematic characterization of cyanobacterial HtpG, which has remained understudied relative to its bacterial and eukaryotic counterparts.

Consistent with other Hsp90 family members, seHtpG exists predominantly as a dimer, as confirmed by size-exclusion chromatography and analytical ultracentrifugation. Dimerization is a hallmark of Hsp90 function, facilitating allosteric communication between domains and client protein maturation. Interestingly, we observed that seHtpG hydrolyzes ATP at a rate of ~1.9 ATP/min, approximately four-fold higher than *E. coli* HtpG under comparable conditions [32]. This elevated activity may reflect adaptations of cyanobacterial HtpG to the dynamic and often stressful environments these organisms inhabit, where rapid protein refolding and stabilization could be advantageous for maintaining photosynthetic efficiency and stress resilience.

The sequence analysis highlights both conserved and unique features of seHtpG. Like other Hsp90 homologs, seHtpG contains the canonical N-terminal ATP-binding domain, middle domain, and C-terminal dimerization domain. However, the absence of the charged linker and MEEVD motif, combined with the presence of a cyanobacteria-specific insertion in the middle domain, points to potential mechanistic divergences from eukaryotic Hsp90s (Figure 1). Such structural distinctions may influence seHtpG environmental adaptation and substrate specificity. Future work should investigate whether the cyanobacteria-specific insertion contributes to client recognition, stability of photosynthetic complexes, or regulation under stress conditions.

Our findings also suggest intriguing evolutionary relationships. Phylogenetic analysis revealed that seHtpG is more closely related to the human mitochondrial Hsp90, Trap1, than to plant chloroplast Hsp90s. This pattern likely reflects the parallel evolutionary histories of cyanobacteria-derived chloroplasts [34] and proteobacteria-derived mitochondria [24,25], hinting at conserved features of organellar proteostasis systems. Such evolutionary parallels raise the possibility that cyanobacterial HtpG may serve as a useful model for exploring the ancestral roles of Hsp90 in endosymbiotic organelles.

Together, these results establish seHtpG as a valuable model for dissecting the roles of Hsp90 in cyanobacterial proteostasis, photosynthesis, and stress response. Beyond basic mechanistic insight, understanding how seHtpG maintains proteome integrity under fluctuating environmental conditions may inform strategies for engineering stress-tolerant cyanobacteria. Such advances could have downstream applications in biotechnology, including biofuel production, carbon capture, and bioremediation. Future studies should aim to (i) identify native client proteins of seHtpG, particularly within photosynthetic and stress-response pathways, (ii) determine high-resolution structures of seHtpG alone and in complex with clients.

In summary, this work provides both a methodological foundation and functional characterization of seHtpG, advancing our understanding of cyanobacterial Hsp90s. By linking conserved chaperone mechanisms to cyanobacteria-specific features, these findings open new avenues for exploring how protein quality control contributes to the ecological resilience and biotechnological potential of cyanobacteria.

## Figures and Tables

**Figure 1 mps-08-00103-f001:**
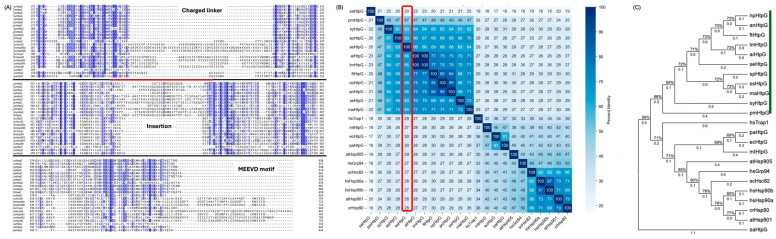
Sequence alignments and phylogenetic analysis of Hsp90 family members across prokaryotic and eukaryotic organisms. (**A**) Partial sequence alignment highlights key differences between *S. elongatus* PCC 7942 HtpG (seHtpG) and other Hsp90 homologs (detailed species name shown in Appendix A). Conserved regions and unique sequence features in seHtpG are indicated. The degree of sequence conservated is highlighted in blue. (**B**) Percent Identity among Hsp90 sequences and the comparison between seHtpG and other homologs are labeled by a red rectangle. (**C**) Phylogenetic tree depicting relationships among representative Hsp90 homologs, illustrating the close evolutionary relationship between cyanobacterial HtpG and mitochondrial Hsp90 (Trap1) from eukaryotes. A branch length shown above branches represents the phylogenetic distance to the original point in terms of sequence divergence and the reliability of branching patterns is indicated by the bootstrap support values on branches. Green lines cover the cyanobacterial strains.

**Figure 2 mps-08-00103-f002:**
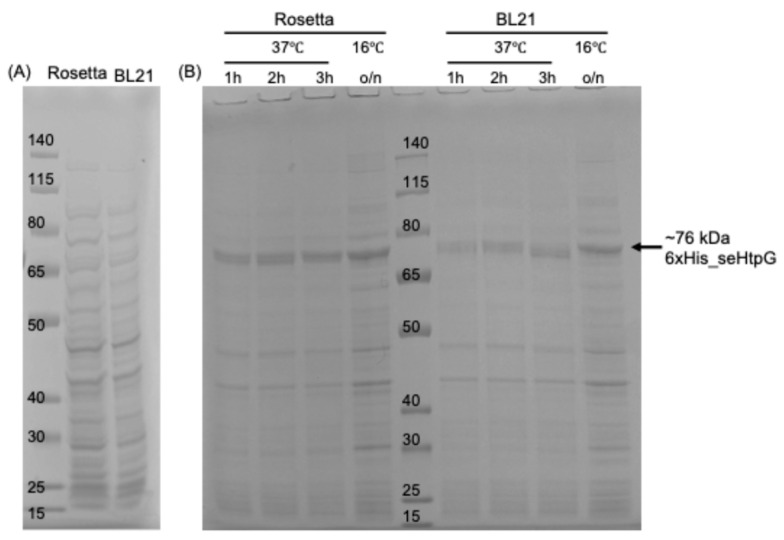
The recombinant overexpression level of seHtpG visualized over *E. coli* cell lysates by SDS-PAGE gel analysis with Coomassie staining. Panel (**A**) shows the lysate of host cell without plasmid transformation. The protein makers (in kDa) were loaded into the left lane of panel (**A**) and the middle lane of panel (**B**).

**Figure 3 mps-08-00103-f003:**
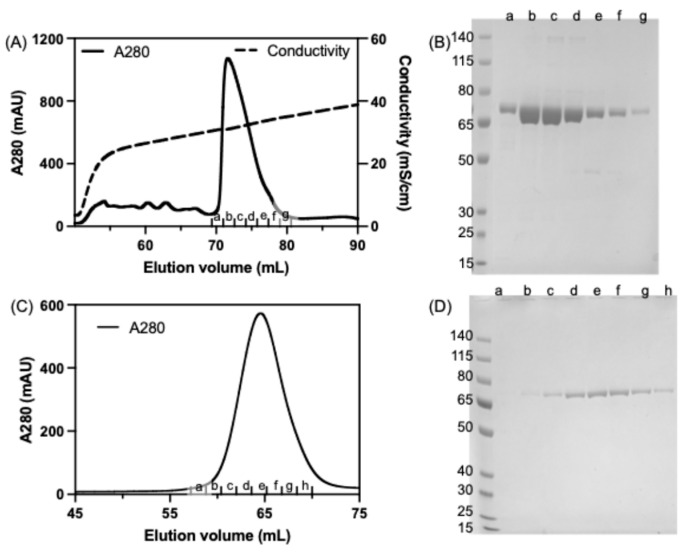
Purification of recombinant seHtpG. (**A**) Elution profile of seHtpG from anion exchange chromatography. (**B**) SDS-PAGE analysis of seHtpG fractions collected from ion exchange chromatography shown in (**A**). The first lane on the left contains the molecular weight marker (in kDa). Lanes a–g correspond to seHtpG sample fractions collected from ion exchange chromatography in (**A**). (**C**) Elution profile of seHtpG from size exclusion chromatography. (**D**) SDS-PAGE analysis of seHtpG fractions collected from size exclusion chromatography shown in (**C**). The first lane on the left contains the molecular weight marker (in kDa). Lanes a–h correspond to seHtpG sample fractions collected from size exclusion chromatography in (**B**).

**Figure 4 mps-08-00103-f004:**
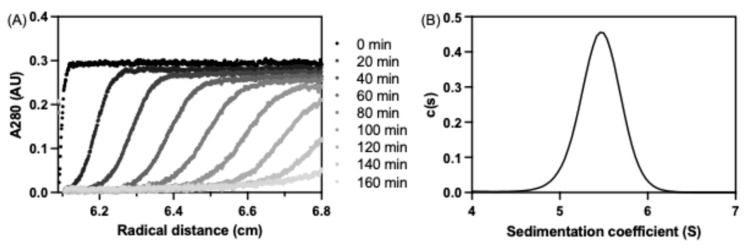
Analytical ultracentrifugation (AUC) analysis of seHtpG. (**A**) Representative sedimentation velocity traces for seHtpG depicted every 20 min for the sake of clarity. Absorbance at 280 nm was monitored for 4 µM seHtpG. (**B**) Sedimentation coefficient distribution for seHtpG, determined from a global fit of the sedimentation velocity traces.

**Figure 5 mps-08-00103-f005:**
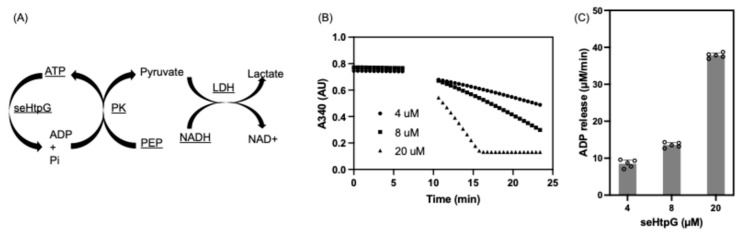
ATP hydrolysis activity of seHtpG measured by NADH based enzymatic coupled assay. (**A**) Schematic illustration of the NADH-coupled enzymatic assay used to monitor ATPase activity. (**B**) Protein concentration dependence of ADP release over time by monitoring NADH absorbance at 340 nm. The reaction mixtures were pre-equilibrated at 37 °C for 7 min before ATP addition. (**C**) ADP release rates at different seHtpG concentrations, determined by fitting the linear regions of the curves in (**B**) after ATP addition. The final ATP hydrolysis rate for seHtpG was calculated as 1.92 ± 0.26 ATP/min by first normalizing the ADP release rate by protein concentration and then averaging across three protein concentrations.

## Data Availability

The original contributions presented in this study are included in the article/Appendix A. Further inquiries can be directed to the corresponding author.

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
