# Peer review of "Recombinant Expression and Purification of the Cyanobacterial Chaperone HtpG from *Synechococcus elongatus* PCC 7942"

_mps, 2025, doi:10.3390/mps8050103_

Round 1
Reviewer 1 Report
Comments and Suggestions for Authors
This paper describes the production, in E. coli, of a cyanobacterial heat shock protein HtpG from Synechococcus elongatus PCC 7942. The authors highlight the importance of this chaperone in the exploitation of this cyanobacterium as a model organism and provide evidence for the sequence, structural and functional homology of HtpG that is shared with other Hsp90-like proteins in both prokaryotes and eukaryotes.
The style is generally clear and reasonably concise, with few minor typos to correct. Additional information in Material and Methods is needed to facilitate protocol reproduction by the reader. The Results and Discussion section would benefit from more objective language and requires more detailed supporting evidence with references. Specific issues are addressed below.
Abstract
Line 8: add quantitative information by a short insertion: ‘…1.8 ATP/min, 3.8-fold faster than that of E. coli HtpG’.
Introduction
Para 3: Define the residue number ranges for the domains and cite (a) reference(s).
Are there any earlier publications describing the recombinant expression of other HtpG/Hsp90-like proteins? Do the authors have specific reasons for selecting HtpG from S. elongatus over other model cyanobacteria to express in E. coli? Is HtpG from other model cyanobacteria more challenging to express in E. coli?
Para4: Add more information, eg. ‘…purification of recombinant seHtpG in E. coli using nickel affinity, anion exchange and size exclusion chromatography.’
Materials and Methods
Para 1, line 1 (missing last letter ‘s’): ‘The sequence alignments…’.
Section 2.1:
Typos and modification: ‘…with abbreviations and Uniprot IDs…’; ‘The phylogenetic tree was constructed using the neighbor-joining…’.
Include a reference for the NJ method and a hyperlink for Mega 11.
Section 2.2:
Specify the software used for codon optimization.
Specify a source for the sequence and structure of the pET28a expression vector.
Section 2.3:
Expand to enable the reader to reproduce the protocol eg. state the culture volumes from which cells were harvested, together with the volumes of buffer used for cell resuspension and lysis. Specify the cell lysis method in more detail, including what instrumentation and settings were used. Provide full information for the chromatography methods, eg. instrumentation, buffer compositions, flowrates, gradient details for IEX etc. Include references as appropriate to avoid repeating some of the details.
If the 4-12% SDS gels are commercially available, the source and size should be cited.
Section 2.4:
Line 3: The pH of HEPES buffer is not adjusted with HCl. Assuming the HEPES was adjusted to pH 7.5 with KOH, change this phrase to ‘…dialyzed overnight at 4°C in buffer containing 40 mM K-HEPES (pH 7.5), 150 mM KCl, 1 mM β-mercaptoethanol.’
Section 2.5:
Cite a reference for this assay and state the source(s) of the enzymes.
The last sentence can be condensed/modified: ‘NADH consumption was monitored by a decrease in absorbance at 340 nm on a Spectramax iD5 plate reader (Molecular Devices, LLC, USA), with ADP release rates calculated using a standard curve over a xxx-xxx range.’
Results and Discussion
Results and discussion (initial letter should be capitalized).
Section 3.1
Para 1:
The final sentence implies that the HtpG orthologs in other organisms exist as more than 1 gene copy. More detail is needed, eg. are the sequences identical and what is the logic behind selecting the specific Uniprot IDs for the alignment if there are other copies present?
Para 2:
Add quantitative information (eg. sequence identities of xx-xx%) to ‘Sequence alignment reveals a high degree of conservation…’.
The abbreviations NTD, MD etc have already been defined in the Introduction. It is acceptable to use only the abbreviations here. Also check later sections for inconsistent use of full wording/abbreviations.
The sequence features described in this section, eg. E391-T408, should be indicated in the figure and legend.
The sentences ‘Mitochondria…these organelles.’ Can be reworded to sound less speculative, eg. ‘It is widely accepted that…’ and should be supported by references to earlier studies.
Figure 1 and legend: Define abbreviations for all organisms in addition to seHtpG or refer to Section 2.1. Provide a more detailed explanation of the blue/pale blue highlighting and add the domains/features described in section 3.1. Explain why alignments start at ~residue 200 instead of the N-terminus.
Panel B, which appears blurred even at higher magnification, needs to be reconstructed. Presumably the % values are sequence identities. This information should be provided in the legend.
The sentences ‘Numbers…divergence.’ Should be reworded to provide a tighter definition of the relationship between branch length and evolutionary distance.
Section 3.2:
Reword to eliminate the phrase ‘…we tested various expression etc…’. A concise summary of what the authors actually did to generate the results is preferable.
Presumably the protein profiles shown in Fig. 2 are derived from the E. coli cell lysates. This information should be given. In addition to the expected molecular mass, more compelling evidence that the ~76 kDa band is seHtpG should be demonstrated, for example by comparison with a non-induced negative control. The identity of this band should also be supported by either (a) in-gel tryptic digestion and analysis by mass spectrometry or (b) immunoblotting or both.
Section 3.3:
Reword to eliminate subjective terminology, eg. ‘…seHtpG is highly negatively charged…’, ‘…low isoelectric point…’, ‘…achieve high purity…’, ‘…high yield…’.
Any previous studies on the structure of Hsp90-like proteins should be cited to support the authors’ hypotheses concerning the surface negative charge and effect of dimerization on the chromatographic behavior of seHtpG.
The statement ‘Purity exceeding 99%...’ is based on visual inspection, therefore assignment of a numerical value (99%) is not appropriate. It would be more acceptable to write ‘On visual inspection of the SEC elution profile, only a single band corresponding to seHtpG was observed, with the absence of contaminating proteins.’
This visual assessment would be more convincing by using a more sensitive staining method than Coomassie Blue, eg. silver or Sypro Ruby.
The yield (~20 mg/L) should be put into context by eg. stating the wet weight of cells and the final volume that this value corresponds with.
Figure 3:
Change the x-axes on panels (A) and (B) to ‘Elution volume (mL)’.
Show SDS-PAGE across the anion exchange elution profile to in addition to the SEC.
Section 3.4:
If the authors were able to gain any information from AUC about the purity of their seHtpG, this information would enhance the impact of the study.
Figure 4A:
Axis fonts should be consistent with other figures. Change the x-axis label to ‘Radial distance (cm)’ and the y-axis to ‘A280’ for consistency with other figures.
Section 3.5:
For readers that are familiar with end-point enzyme assays, it would be helpful to briefly explain how the coupled ATPase assay works and why it is advantageous.
The authors report results for the mean of ATPase assays at 3 seHtpG concentrations. A more statistically valid approach would be to assay 3 replicates at each of the 3 concentrations and, as the authors have reported, expressing the results as rate of ADP production as a function of seHtpG concentration. Figure 5 would additionally benefit from the inclusion of error bars and, in view of the clear difference in ATPase activity between seHtpG and the orthologs from other organisms, make the conclusion more convincing.
Do the authors have a hypothesis to explain why the ATP hydrolysis rate of seHtpG is substantially higher than the other orthologs?
To provide additional information for the reader, ATPase assays should be performed to assess the stability of ATPase activity post purification, alongside SDS-PAGE profiles to detect protein degradation. Recommended storage conditions for purified seHtpG would be a valuable resource.
Author Response
Responses to Reviewer #1:
This paper describes the production, in E. coli, of a cyanobacterial heat shock protein HtpG from Synechococcus elongatus PCC 7942. The authors highlight the importance of this chaperone in the exploitation of this cyanobacterium as a model organism and provide evidence for the sequence, structural and functional homology of HtpG that is shared with other Hsp90-like proteins in both prokaryotes and eukaryotes. The style is generally clear and reasonably concise, with few minor typos to correct. Additional information in Material and Methods is needed to facilitate protocol reproduction by the reader. The Results and Discussion section would benefit from more objective language and requires more detailed supporting evidence with references. Specific issues are addressed below.
We thank the reviewer for carefully reading our manuscript and for providing an overall favorable evaluation. We have addressed the concerns point by point and revised the manuscript accordingly, based on the reviewer’s thoughtful suggestions.
Abstract
Line 8: add quantitative information by a short insertion: ‘…1.8 ATP/min, 3.8-fold faster than that of E. coli HtpG’.
Thank you for the suggestion. We have added the quantitative information to the Abstract.
Introduction
Para 3: Define the residue number ranges for the domains and cite (a) reference(s).
Thank you for the suggestions. We have defined the residue numbers for each domain in seHtpG based on its sequence alignment with E. coli HtpG and the established domain definitions of E. coli HtpG.
Are there any earlier publications describing the recombinant expression of other HtpG/Hsp90-like proteins?
Yes, earlier studies have described the recombinant expression and purification of E. coli HtpG and other Hsp90 homologs from yeast and human. However, in most cases these were presented only as brief methodological notes. In contrast, our work provides a detailed description of the expression and purification procedure. In addition, we are extending this work to cyanobacterial HtpG. To highlight this distinction, we have added a sentence to the introduction along with appropriate references.
Do the authors have specific reasons for selecting HtpG from S. elongatus over other model cyanobacteria to express in E. coli? Is HtpG from other model cyanobacteria more challenging to express in E. coli?
Synechococcus elongatus is a widely used cyanobacterial model organism for research on prokaryotic photosynthesis and circadian rhythms. It was selected as a model system due to its relatively simple genome and rapid growth rate. We also attempted to purify HtpG from another well-established unicellular model, Synechococcus sp. PCC 7002, but the expression and purification yields were substantially lower. Therefore, we chose to focus our efforts on S. elongatus.
Para4: Add more information, eg. ‘…purification of recombinant seHtpG in E. coli using nickel affinity, anion exchange and size exclusion chromatography.’
Thank you for the suggestion. All chromatography steps used in the purification process are now explicitly described in the Introduction.
Materials and Methods
Para 1, line 1 (missing last letter ‘s’): ‘The sequence alignments…’.
Corrected.
Section 2.1:
Typos and modification: ‘…with abbreviations and Uniprot IDs…’; ‘The phylogenetic tree was constructed using the neighbor-joining…’.
Include a reference for the NJ method and a hyperlink for Mega 11.
Thanks for the detailed comments. All typos have been corrected. The reference for NJ algorithm and Mega website was added.
Section 2.2:
Specify the software used for codon optimization.
Specify a source for the sequence and structure of the pET28a expression vector.
Codon optimization, cloning into the pET28a vector, and construct preparation were performed by Twist Bioscience, as now described in Section 2.2.
Section 2.3:
Expand to enable the reader to reproduce the protocol eg. state the culture volumes from which cells were harvested, together with the volumes of buffer used for cell resuspension and lysis. Specify the cell lysis method in more detail, including what instrumentation and settings were used. Provide full information for the chromatography methods, eg. instrumentation, buffer compositions, flowrates, gradient details for IEX etc. Include references as appropriate to avoid repeating some of the details.
If the 4-12% SDS gels are commercially available, the source and size should be cited.
Thanks for the comments. We have expanded the expression and purification protocols to incorporate all the details requested by the reviewers.
Section 2.4:
Line 3: The pH of HEPES buffer is not adjusted with HCl. Assuming the HEPES was adjusted to pH 7.5 with KOH, change this phrase to ‘…dialyzed overnight at 4°C in buffer containing 40 mM K-HEPES (pH 7.5), 150 mM KCl, 1 mM β-mercaptoethanol.’
Thanks for catching the typo. It is KOH that was used to adjust the pH of HEPES to 7.5 and the typo has been corrected.
Section 2.5:
Cite a reference for this assay and state the source(s) of the enzymes.
The last sentence can be condensed/modified: ‘NADH consumption was monitored by a decrease in absorbance at 340 nm on a Spectramax iD5 plate reader (Molecular Devices, LLC, USA), with ADP release rates calculated using a standard curve over a xxx-xxx range.’
We have added a reference for the enzymatic coupled ATPase assay. The last sentence is revised as suggested.
Results and Discussion
Results and discussion (initial letter should be capitalized).
Corrected
Section 3.1
Para 1:
The final sentence implies that the HtpG orthologs in other organisms exist as more than 1 gene copy. More detail is needed, eg. are the sequences identical and what is the logic behind selecting the specific Uniprot IDs for the alignment if there are other copies present?
The phrase “single copy” in the final sentence of paragraph 1 was intended to emphasize that S. elongatus PCC 7942, as a prokaryote, contains only one copy of the hsp90 gene, in contrast to eukaryotic organisms, which typically harbor multiple paralogs. For example, humans have four distinct Hsp90 genes: two cytosolic isoforms and two organelle-specific isoforms, one localized to the endoplasmic reticulum and the other to mitochondria. All four human Hsp90 sequences were included in our sequence alignments. To clarify this point, we have added a sentence to the manuscript.
Para 2:
Add quantitative information (eg. sequence identities of xx-xx%) to ‘Sequence alignment reveals a high degree of conservation…’.
Good point. To further clarify this comparison and make it quantitative, we have added a new figure illustrating the pairwise sequence identities among the representative Hsp90 sequences.
The abbreviations NTD, MD etc have already been defined in the Introduction. It is acceptable to use only the abbreviations here. Also check later sections for inconsistent use of full wording/abbreviations.
Thank you for the suggestion. We have carefully reviewed the manuscript and confirmed that abbreviations are used consistently throughout, except where they are defined upon first use.
The sequence features described in this section, eg. E391-T408, should be indicated in the figure and legend.
Thank you for the suggestion. We have highlighted the three unique sequences in Figure 1 for clarity.
The sentences ‘Mitochondria…these organelles.’ Can be reworded to sound less speculative, eg. ‘It is widely accepted that…’ and should be supported by references to earlier studies.
Thank you for the suggestion. The sentence has been revised with the appropriate references.
Figure 1 and legend: Define abbreviations for all organisms in addition to seHtpG or refer to Section 2.1. Provide a more detailed explanation of the blue/pale blue highlighting and add the domains/features described in section 3.1. Explain why alignments start at ~residue 200 instead of the N-terminus.
Panel B, which appears blurred even at higher magnification, needs to be reconstructed. Presumably the % values are sequence identities. This information should be provided in the legend.
The sentences ‘Numbers…divergence.’ Should be reworded to provide a tighter definition of the relationship between branch length and evolutionary distance.
The caption of Figure 1 has been revised to better explain the abbreviations and color highlights. Sequence alignments were performed using full-length sequences; however, only the regions showing significant divergence are displayed in Figure 1A. To avoid confusion, the complete full-length alignments have been provided in the supplementary materials. Panel B of Figure 1 has also been updated to a higher resolution. In addition, the final sentence has been reworded to more clearly explain branch length, bootstrap values, and their relationship.
Section 3.2:
Reword to eliminate the phrase ‘…we tested various expression etc…’. A concise summary of what the authors actually did to generate the results is preferable.
Thanks for the suggestion. The first sentence of section 3.2 has been revised to be more specific.
Presumably the protein profiles shown in Fig. 2 are derived from the E. coli cell lysates. This information should be given. In addition to the expected molecular mass, more compelling evidence that the ~76 kDa band is seHtpG should be demonstrated, for example by comparison with a non-induced negative control. The identity of this band should also be supported by either (a) in-gel tryptic digestion and analysis by mass spectrometry or (b) immunoblotting or both.
The protein profiles are derived from bacterial cell lysates, and this information has now been added to the text and figure captions. While we did not perform mass spectrometry or Western blotting to confirm the identity of seHtpG, the distinct band shift observed after overnight 6×His-tag cleavage verified the expression of His-tagged seHtpG. Furthermore, downstream confirmation of dimer formation by AUC and validation of ATPase activity strongly support that the purified product is indeed seHtpG.
Section 3.3:
Reword to eliminate subjective terminology, eg. ‘…seHtpG is highly negatively charged…’, ‘…low isoelectric point…’, ‘…achieve high purity…’, ‘…high yield…’.
Thanks for the suggestion. The sentence has been revised to remove the subjective terminology.
Any previous studies on the structure of Hsp90-like proteins should be cited to support the authors’ hypotheses concerning the surface negative charge and effect of dimerization on the chromatographic behavior of seHtpG.
The negative surface charge of seHtpG is supported by its binding to the Mono Q column, a strong anion exchanger. seHtpG eluted at a salt concentration of ~300 mM KCl, which is considerably higher than expected for a monomeric protein with a pI of 4.9, suggesting dimer formation. This dimerization was further confirmed by AUC, as described later in the manuscript.
The statement ‘Purity exceeding 99%...’ is based on visual inspection, therefore assignment of a numerical value (99%) is not appropriate. It would be more acceptable to write ‘On visual inspection of the SEC elution profile, only a single band corresponding to seHtpG was observed, with the absence of contaminating proteins.’ This visual assessment would be more convincing by using a more sensitive staining method than Coomassie Blue, eg. silver or Sypro Ruby.
Thank you for the suggestion. We have removed the statement claiming 99% purity and instead now describe the purity assessment based on the SEC elution profile and SDS-PAGE analysis. Although we did not perform silver staining or Sypro Ruby staining in this case, we believe that the SEC elution profile together with SDS-PAGE using Coomassie Blue provides sufficient evidence of sample purity.
The yield (~20 mg/L) should be put into context by eg. stating the wet weight of cells and the final volume that this value corresponds with.
The protein yield was calculated based on a 1 L bacterial culture, and this information has now been included in the text. Protein yield is conventionally reported in units of mg/L, as wet cell weight or final culture volume are not reliable indicators. Total cell mass can vary between batches due to differences in OD measurements and harvest timing, while the average protein expressed per cell, although difficult to measure, would be a more meaningful metric. Therefore, we report the yield in mg/L based on the final purified protein measurements.
Figure 3:
Change the x-axes on panels (A) and (B) to ‘Elution volume (mL)’.
Show SDS-PAGE across the anion exchange elution profile to in addition to the SEC.
Thank you for the suggestion. The x-axes have been revised and the SDS-PAGE for samples from ion exchange elution has been provided and shown as Figure 3B. The previous Figure 3B and C of size exclusion profile are Figure 3 C and D in current version.
Section 3.4:
If the authors were able to gain any information from AUC about the purity of their seHtpG, this information would enhance the impact of the study.
Our AUC system detects absorbance only at 280 nm, which requires relatively high protein concentrations (absorbance > 0.3 with a 1.2 cm pathlength) to obtain sufficient signal for data analysis. Under these conditions, minor impurities are unlikely to appear as distinct peaks in the AUC profiles. Therefore, we continue to rely on SEC and SDS-PAGE for assessing sample purity, which we believe provide sufficient validation.
Figure 4A:
Axis fonts should be consistent with other figures. Change the x-axis label to ‘Radial distance (cm)’ and the y-axis to ‘A280’ for consistency with other figures.
Thanks for the comment. The Figure 4 has been updated.
Section 3.5:
For readers that are familiar with end-point enzyme assays, it would be helpful to briefly explain how the coupled ATPase assay works and why it is advantageous.
Thank you for the suggestion. The detailed reactions involved in the ATPase assay are supplemented to section 3.5 and a schematic presentation for reactions are provided to Figure 5.
The authors report results for the mean of ATPase assays at 3 seHtpG concentrations. A more statistically valid approach would be to assay 3 replicates at each of the 3 concentrations and, as the authors have reported, expressing the results as rate of ADP production as a function of seHtpG concentration. Figure 5 would additionally benefit from the inclusion of error bars and, in view of the clear difference in ATPase activity between seHtpG and the orthologs from other organisms, make the conclusion more convincing.
In the revised manuscript, we now included all 5 replicates for each concentration and the values of ATP hydrolysis rate in unit of per seHtpG monomer in Figure 5.
Do the authors have a hypothesis to explain why the ATP hydrolysis rate of seHtpG is substantially higher than the other orthologs?
We do not have a concrete explanation for why seHtpG hydrolyzes ATP faster than other orthologs. One possibility is that it may be related to the organism’s natural environment, which experiences frequent temperature fluctuations. However, as we do not currently have sufficient evidence to support this hypothesis, we intentionally did not include such speculation in the manuscript.
To provide additional information for the reader, ATPase assays should be performed to assess the stability of ATPase activity post purification, alongside SDS-PAGE profiles to detect protein degradation. Recommended storage conditions for purified seHtpG would be a valuable resource.
We repeated the ATPase assays five times at each protein concentration, as shown in Figure 5. Individual data points, averages, and standard deviations are provided. No significant differences were observed in ATPase activity before and after storage at –80 °C. The storage conditions are described in the Methods section.

Reviewer 2 Report
Comments and Suggestions for Authors
Review mps-3826706
I congratulate authors on their research effort. Although the data produced is solid, authors fell short on some aspects of the manuscript, mainly in the Results and Discussion section being so condensed. I believe that further context from the literature and maybe adding a few more strains in their phylogenetic analysis can enhance the manuscript’s quality and robustness
Abstract
Excellent presentation of the work, with good amount of background information and data produced by authors
Introduction
Page 2 Line 15 – it would be interesting if authors could cite here the % of identity and coverage between seHtpG and Hsp90
All the details describing seHtpG included in the Introduction sound like a result section. Maybe authors could describe the general structure of HtpGs and indicate whether the cyanobacterial counterpart is similar to that (this information is still super relevant and it fits perfectly in page 3, line 9 of section 3)
The strain used by authors, PCC 7942, was isolated from a freshwater sample originally from California (USA). Authors could benefit from expanding in their introduction the cosmopolitan nature of cyanobacteria and how they produce several compounds to survive from hot tropical to cold Antarctic weather, meaning that HtpG could function as an adaptive trait for many functions in such a diverse group of organisms, acting from a cold response to oxidative stress defense. Also, are authors aware of the work from Sato et al (2010) that investigated exactly the same Hsp90 from S. elongatus PCC7942? (https://doi.org/10.1111/j.1365-2958.2010.07139.x ). I was only briefly cited in the introduction but maybe it could be further cited in the discussion section by comparing structural data. Also, Tanaka and Nakamoto already showed evidence of a thermo-stress defense mechanism related with HtpG from PCC7942 in vivo (https://doi.org/10.1016/S0014-5793(99)01134-5 )
Material and Methods
Page 3 Line 3 – authors could inform the parameters used for sonication and centrifugation (time, intensity, etc.)
Item 2.5 – no positive or negative controls were used in the ATPase assay? If yes, please describe it here. Positive control would mostly like be an ADP + assay reagents mix while Negative control could have been ATP without the enzyme to evaluate a stable absorbance over time. If no control was used, authors should describe why they chose not to include those
Results and Discussion
Page 3 Line 6 – authors cited the whole genome with this NCBI accession code. Please cite the htpg gene used. (Maybe GenBank: AB010001.1 ?). Also, if the idea is to investigate the chaperone presence in cyanobacteria, why did the authors not included more sequences from cyano? A quick BLAST of the HtpG sequence returns 21 results from other strains (mostly synechococcus) that authors could have included. Many other cyano have HtpG sequences, such as Prochlorococcus marinus str. MIT 9515 (WP_011820292.1), Fischerella thermalis PCC 7521 (WP_009459780.1), Microcystis aeruginosa NIES-88 (WP_061433409.1) and Nostoc linckia z6 (WP_099071801.1).
Page 3 Line 8 – authors refer to other cyanobacterial species without naming them. This is kind of confusing, since the figure does not further elucidate this matter.
Page 3 Line 14 – “cyanobacterial HtpG contains a unique 10-30 residue insertion” if authors are referring to the Synechococcus sequence, please specify. Otherwise, it is quite bold to describe the gene structure of an entire phylum without using an expressive sample number. The same goes for the passage “its presence is conserved across cyanobacteria”. Based on how many genera was this conclusion made?
The difference pointed out by authors could be an assembly bias. Without a significant sample size, it is very difficult to confidently claim something.
The phylogenetic closeness of seHtpG to human Hsp90 is definitely a construct generated by the lack of diversity in the phylogenetic analysis. I recommend adding at least 20 + sequences from across Bacteria, including more cyanobacteria, proteo and actino and even search for it in Archaea.
The common shared history between chloroplast and mitochondria is an enormous debate in literature. While trying to add data to that debate with a single gene from only two species seems like a good idea, authors end up having a very superficial discussion on the topic, deviating from their goal of describing the cyanobacterial HTpG gene. Mitochondria might have evolved from α-proteobacteria, probably closely related to modern Rickettsia-like bacteria. Chloroplasts evolved later from cyanobacteria, a completely different bacterial lineage. It seems that an ancient archaeal host cell engulfed (but didn’t digest) an α-proteobacterium, generating a symbiosis that became the first mitochondrion. Much later, in a eukaryotic lineage that already had mitochondria, the cell engulfed a cyanobacterium, originating the first chloroplast (in the ancestor of plants and algae). So, while they share the same mechanism of origin (primary endosymbiosis), these organelles have different bacterial ancestors.
Figure 1 and everywhere else in the manuscript = authors should double check that strains names (genus and species) are italicized throughout the text and in figure captions
Page 4 – authors state that several expression and induction conditions were tested. It would be great if these data were also shared, either by a text or a table (maybe supplementary material?)
Page 4 – “High protein yield was achieved” of how much? Please add numbers here. In this section, no discussion is provided, comparing the results obtained in the study against other cyanobacterial genes successfully expressed in E.coli. The same happens to section 3.3, where data is only presented without context from the literature. If fusing Results and Discussion is going to hurt the quality of the manuscript, I suggest doing them separately as suggested in the author guideline of MPs: The structure should include an Abstract, Keywords, Introduction, Materials and Methods, Results, Discussion, and Conclusions (optional) sections.
I do believe that the discussion pieces provided are too little and authors would benefit from either bringing further information from the literature for comparison or splitting results and discussion to focus on providing detailed contextualization to their data.
Author Response
We sincerely thank both reviewers for taking the time to evaluate our manuscript. We greatly appreciate the constructive comments and suggestions, which have been very helpful in improving the quality and clarity of our work. Below, we provide detailed, point-by-point responses to each comment. The reviewers’ comments are highlighted in blue, and our responses are provided in black. The corresponding revisions and corrections have been incorporated into the revised manuscript, with changes highlighted (tracked) in the re-submitted files.
Response to Review #2
I congratulate authors on their research effort. Although the data produced is solid, authors fell short on some aspects of the manuscript, mainly in the Results and Discussion section being so condensed. I believe that further context from the literature and maybe adding a few more strains in their phylogenetic analysis can enhance the manuscript’s quality and robustness
We thank the reviewer for carefully reading our manuscript and for providing an overall favorable evaluation. We have addressed the concerns point by point and revised the manuscript accordingly, based on the reviewer’s thoughtful suggestions.
Abstract
Excellent presentation of the work, with good amount of background information and data produced by authors
Thanks for your favorable comments.
Introduction
Page 2 Line 15 – it would be interesting if authors could cite here the % of identity and coverage between seHtpG and Hsp90
Sequence identity between seHtpG and other Hsp90 homologs has now been included in the text. In addition, we have added a new figure presenting the pairwise sequence identity between seHtpG and other Hsp90 homologs.
All the details describing seHtpG included in the Introduction sound like a result section. Maybe authors could describe the general structure of HtpGs and indicate whether the cyanobacterial counterpart is similar to that (this information is still super relevant and it fits perfectly in page 3, line 9 of section 3)
Thank you for the suggestion. We have moved the content related to the sequence and structure of seHtpG to the Results section.
The strain used by authors, PCC 7942, was isolated from a freshwater sample originally from California (USA). Authors could benefit from expanding in their introduction the cosmopolitan nature of cyanobacteria and how they produce several compounds to survive from hot tropical to cold Antarctic weather, meaning that HtpG could function as an adaptive trait for many functions in such a diverse group of organisms, acting from a cold response to oxidative stress defense. Also, are authors aware of the work from Sato et al (2010) that investigated exactly the same Hsp90 from S. elongatus PCC7942? (https://doi.org/10.1111/j.1365-2958.2010.07139.x). I was only briefly cited in the introduction but maybe it could be further cited in the discussion section by comparing structural data. Also, Tanaka and Nakamoto already showed evidence of a thermo-stress defense mechanism related with HtpG from PCC7942 in vivo (https://doi.org/10.1016/S0014-5793(99)01134-5 )
Thank you for the suggestion. We have added more context in the Introduction regarding cyanobacterial environmental adaptation and its relevance to chaperones and HtpG. In addition, the work by Sato and Tanaka has been further discussed in the Results and Discussion.
Material and Methods
Page 3 Line 3 – authors could inform the parameters used for sonication and centrifugation (time, intensity, etc.)
The details about sonication and centrifugation have been provided in section 2.3.
Item 2.5 – no positive or negative controls were used in the ATPase assay? If yes, please describe it here. Positive control would mostly like be an ADP + assay reagents mix while Negative control could have been ATP without the enzyme to evaluate a stable absorbance over time. If no control was used, authors should describe why they chose not to include those
Thank you for the suggestion. We have included a seHtpG-free condition as the negative control and clarified this in Section 2.5. The positive control referenced by the reviewer was already performed and was used to generate the ATPase standard curve, which allowed us to convert relative ATPase rates into absolute ATPase activities.
Results and Discussion
Page 3 Line 6 – authors cited the whole genome with this NCBI accession code. Please cite the htpg gene used. (Maybe GenBank: AB010001.1 ?). Also, if the idea is to investigate the chaperone presence in cyanobacteria, why did the authors not included more sequences from cyano? A quick BLAST of the HtpG sequence returns 21 results from other strains (mostly synechococcus) that authors could have included. Many other cyano have HtpG sequences, such as Prochlorococcus marinus str. MIT 9515 (WP_011820292.1), Fischerella thermalis PCC 7521 (WP_009459780.1), Microcystis aeruginosa NIES-88 (WP_061433409.1) and Nostoc linckia z6 (WP_099071801.1).
Thank you for the suggestion. We have replaced the whole-genome accession with the GenBank number for seHtpG. To support our conclusion regarding the cyanobacteria-specific insertion, we performed sequence alignments across more than 20 cyanobacterial species, representing freshwater and marine strains, filamentous and unicellular forms, as well as both toxin-producing and non-toxic species. Figure 1 has been updated in the revised manuscript to include additional HtpG sequences.
Page 3 Line 8 – authors refer to other cyanobacterial species without naming them. This is kind of confusing, since the figure does not further elucidate this matter.
We have revised the sentence to “Sequence alignment reveals a high degree of conservation, particularly within the cyanobacterial species. The pairwise sequence identities among the representative Hsp90 sequences is shown in Fig. 1.” The name of other cyanobacterial species is given in Section 2.1.
Page 3 Line 14 – “cyanobacterial HtpG contains a unique 10-30 residue insertion” if authors are referring to the Synechococcus sequence, please specify. Otherwise, it is quite bold to describe the gene structure of an entire phylum without using an expressive sample number. The same goes for the passage “its presence is conserved across cyanobacteria”. Based on how many genera was this conclusion made? The difference pointed out by authors could be an assembly bias. Without a significant sample size, it is very difficult to confidently claim something.
The unique insertion is indeed a striking observation to us as well. For all the cyanobacterial sequences examined to date (>20), our conclusion still holds. However, we agree with the reviewer that the original language may have been too strong given the limited sampling size and potential assembly biases. There may well be exceptions in nature. Accordingly, we have toned down the language in the revised manuscript.
The phylogenetic closeness of seHtpG to human Hsp90 is definitely a construct generated by the lack of diversity in the phylogenetic analysis. I recommend adding at least 20 + sequences from across Bacteria, including more cyanobacteria, proteo and actino and even search for it in Archaea. The common shared history between chloroplast and mitochondria is an enormous debate in literature. While trying to add data to that debate with a single gene from only two species seems like a good idea, authors end up having a very superficial discussion on the topic, deviating from their goal of describing the cyanobacterial HTpG gene. Mitochondria might have evolved from α-proteobacteria, probably closely related to modern Rickettsia-like bacteria. Chloroplasts evolved later from cyanobacteria, a completely different bacterial lineage. It seems that an ancient archaeal host cell engulfed (but didn’t digest) an α-proteobacterium, generating a symbiosis that became the first mitochondrion. Much later, in a eukaryotic lineage that already had mitochondria, the cell engulfed a cyanobacterium, originating the first chloroplast (in the ancestor of plants and algae). So, while they share the same mechanism of origin (primary endosymbiosis), these organelles have different bacterial ancestors.
Thank you for the clarification and suggestions. In Section 3.1 and Figure 1, we have incorporated these by including additional cyanobacterial species as well as representatives from archaea, actinomycetota, and Pseudomonas. Even with the expanded dataset, our phylogenetic analysis still shows that human mitochondrial Trap1 is most closely related to cyanobacterial HtpG. More reference about mitochondria and chloroplast are provided as well.
Figure 1 and everywhere else in the manuscript = authors should double check that strains names (genus and species) are italicized throughout the text and in figure captions
We confirmed that all strain names are correct and italicized.
Page 4 – authors state that several expression and induction conditions were tested. It would be great if these data were also shared, either by a text or a table (maybe supplementary material?)
The expression host, induction temperature and duration were tested for seHtpG expression and the results are visualized by SDS-PAGE on Figure 2.
Page 4 – “High protein yield was achieved” of how much? Please add numbers here.
We did not quantify the protein yield at this initial stage; it was evaluated primarily by visual inspection of gel band intensity. Another reason we didn’t quantify it is that there isn’t much difference between the compared conditions. The targeted seHtpG stands out as the overexpress protein in all cases. Under the optimized conditions, the final yield after purification was approximately 20 mg/L, which provides a more accurate and meaningful estimate based on absorbance measurements of the purified protein.
In this section, no discussion is provided, comparing the results obtained in the study against other cyanobacterial genes successfully expressed in E.coli. The same happens to section 3.3, where data is only presented without context from the literature. If fusing Results and Discussion is going to hurt the quality of the manuscript, I suggest doing them separately as suggested in the author guideline of MPs: The structure should include an Abstract, Keywords, Introduction, Materials and Methods, Results, Discussion, and Conclusions (optional) sections. I do believe that the discussion pieces provided are too little and authors would benefit from either bringing further information from the literature for comparison or splitting results and discussion to focus on providing detailed contextualization to their data.
Thank you for the suggestion. The current manuscript is focused on reporting an expression and purification protocol for seHtpG, with only limited discussion of its biochemical and structural properties. We have demonstrated that seHtpG forms a homodimer by AUC and confirmed its activity through ATPase assays. However, we have intentionally restricted the scope of this work to expression and purification. Follow-up studies are in progress to provide a more detailed biochemical and structural characterization of seHtpG. If acceptable to the editor, we would prefer to maintain the current manuscript format, with Results and Discussion presented together.

Round 2
Reviewer 1 Report
Comments and Suggestions for Authors
The authors have diligently addressed all of issues raised in my reviewer's report. I am therefore please to recommend that this paper can be published in its present form.
Author Response
We're so glad that our revision meets your expectation. Thanks for your time again.
Reviewer 2 Report
Comments and Suggestions for Authors
Authors have evaluated my previous report and incorporated the suggestions they deemed as worthy. I do believe that the new version is improved. However, I still believe that a version with a separate Discussion section would be stronger.
Author Response
[Authors have evaluated my previous report and incorporated the suggestions they deemed as worthy. I do believe that the new version is improved. However, I still believe that a version with a separate Discussion section would be stronger. ]
Thanks for accepting our revision work. We really appreciate your insist on a separate discussion that help us further improve our manuscript. A separate Discussion section has been prepared and highlighted in the new submission.